# ClipGrader: Leveraging Vision-Language Models for Robust Label Quality Assessment in Object Detection

## Abstract

High-quality annotations are essential for object detection models, but ensuring label accuracy — especially for bounding boxes — remains both challenging and costly. This paper introduces ClipGrader, a novel approach that leverages vision-language models to automatically assess the accuracy of bounding box annotations. By adapting CLIP (Contrastive Language-Image Pre-training) to evaluate both class label correctness and spatial precision of bounding box, ClipGrader offers an effective solution for grading object detection labels. Tested on modified object detection datasets with artificially disturbed bounding boxes, ClipGrader achieves 91% accuracy on COCO with a 1.8% false positive rate. Moreover, it maintains 87% accuracy with a 2.1% false positive rate when trained on just 10% of the COCO data. ClipGrader also scales effectively to larger datasets such as LVIS, achieving 79% accuracy across 1,203 classes. Our experiments demonstrate ClipGrader's ability to identify errors in existing COCO annotations, highlighting its potential for dataset refinement. When integrated into a semi-supervised object detection (SSOD) model, ClipGrader readily improves the pseudo label quality, helping achieve higher mAP (mean Average Precision) throughout the training process. ClipGrader thus provides a scalable AI-assisted tool for enhancing annotation quality control and verifying annotations in large-scale object detection datasets.

## 1 Introduction

The performance of machine learning models is heavily dependent on the quality of training data. In object detection, a fundamental task in computer vision, the accuracy of annotations which encompasses both the correctness of the class label and spatial precision of the bounding box is crucial. However, curating high-quality object detection datasets is a significant challenge due to the time-consuming and expensive nature of manual annotation processes, not to mention the inevitability of errors creeping in (Kuznetsova et al., 2018; Vondrick et al., 2013). Existing approaches to data collection and annotation, such as crowd sourcing, web scraping, or AI-generated labels, often introduce noise and inconsistencies, potentially compromising model performance (Papadopoulos et al., 2016; Northcutt et al., 2021; Zare & Yazdi, 2022). With the increasing complexity and scale of datasets, traditional methods of quality control such as manual reviews or simple heuristics struggle to meet the demand. Even widely-used datasets such as COCO (Lin et al., 2014) are subject to label errors, as revealed in prior research (Wang et al., 2022; Chachuła et al., 2023). This highlights the need for automated systems that can efficiently identify and flag noisy labels for review.

In recent years, foundation models have demonstrated remarkable capabilities across various domains (Bommasani et al., 2022). One such model, CLIP (Contrastive Language-Image Pre-training) (Radford et al., 2021), has shown impressive zero-shot performance on a wide range of visual classification tasks. CLIP's ability to understand both visual and textual information makes it a promising candidate to assess the quality of object detection labels. We hypothesize that the task of grading label quality is inherently simpler than the task of object detection itself. This allows our grader to achieve high performance with relatively less data and computational resources compared to training an accurate object detection model. This principle is similar to that observed in reward models (Ouyang et al., 2022) of Reinforcement Learning from Human Feedback (RLHF).

However, CLIP's original architecture was designed for associating images with textual descriptions. Hence it excels in many classification tasks, but does not inherently understand the concept of markers (bounding boxes) added to an image or their quality. Our key innovation lies in developing a training strategy that successfully fine-tunes CLIP to understand what constitutes a "good" bounding box and make accurate judgments about label quality. This involves teaching the model to assess not only the correctness of the class label but also the bounding box's location and tightness. Notably, our results demonstrates that CLIP can learn and understand the concept of "markers" on images, potentially opening new avenues for visual reasoning tasks (Wu et al., 2017). The key contributions of this paper are as follows:

- We introduce *ClipGrader*, a novel framework that leverages vision-language models to automatically assess both class label correctness and bounding box spatial accuracy for object detection annotations.

- We demonstrate that our training strategy enables CLIP to understand and evaluate bounding box "markers" on images — potentially introducing a new capability for vision-language models and expanding their applications.

- Our results highlight ClipGrader's high accuracy, robustness and data efficiency across various setups. It maintains strong performance even with very limited labeled data, demonstrating its effectiveness in providing effective quality control for object detection data curation and improving semi-supervised object detection (SSOD) models (Wang et al., 2023) through pseudo label screening.

Interestingly, during our error analysis, we observed cases where ClipGrader's inference results turned out to be more accurate than the ground truth annotations in COCO (see Appendix B), despite ClipGrader being trained with COCO dataset. This phenomenon could be attributed to human errors and inconsistencies in labeling or ambiguities in class definitions (Ma et al., 2022) and underscores the potential of our approach to contribute to the refinement and improvement of even widely-used benchmarks in the field.

## 2 RELATED WORK

**CLIP and its adaptations for related vision tasks** CLIP (Contrastive Language-Image Pre-training) (Radford et al., 2021) has emerged as a powerful vision-language model, demonstrating impressive zero-shot classification capabilities across a wide range of visual tasks. Its architecture, combining a vision encoder and a text encoder trained on a large-scale dataset of image-text pairs, allows it to understand and relate visual and textual information effectively. Several works have attempted to adapt CLIP for object detection and segmentation tasks. Notable examples include RegionClip(Zhong et al., 2021), CLIPSeg (Lüddecke & Ecker, 2022) and GLIP (Li et al., 2022). However, these works focus on direct visual tasks like segmentation or detection, whereas our approach is the first to leverage CLIP as a grader to assess the quality of bounding box annotations. By using CLIP's multimodal understanding, we evaluate both class label correctness and spatial accuracy, introducing a novel application that shifts CLIP's role from a detector to a quality assurance tool for object detection datasets.

**Label quality assessment** Label quality assessment has been an ongoing challenge in machine learning, particularly for large-scale datasets. Previous works have explored various approaches, including confidence learning (Northcutt et al., 2022; Chachuła et al., 2023) and learning with noisy labels (Song et al., 2022; Wang et al., 2022). Our work contributes to this field by introducing a novel foundation model-based approach specifically tailored for object detection tasks, focusing on both class label correctness and bounding box quality. Also, ClipGrader itself is not a object detection model. It could be trained more data efficiently, such that it would be made available early in the data curation process with only very few seed label data available.

**Semi-Supervised Learning in Object Detection** Semi-supervised learning approaches, such as ConsistentTeacher (Wang et al., 2023), have shown great promise by generating pseudo-labels from large amounts of unlabeled data. These methods (Xu et al., 2021; Sohn et al., 2020) typically rely on internal confidence measures or consistency between pseudo-labels to evaluate these self-generated pseudo-labels, but they may struggle to accurately assess the quality of the AI-generated annotations. ClipGrader complements these approaches by providing an external, language-grounded assessment

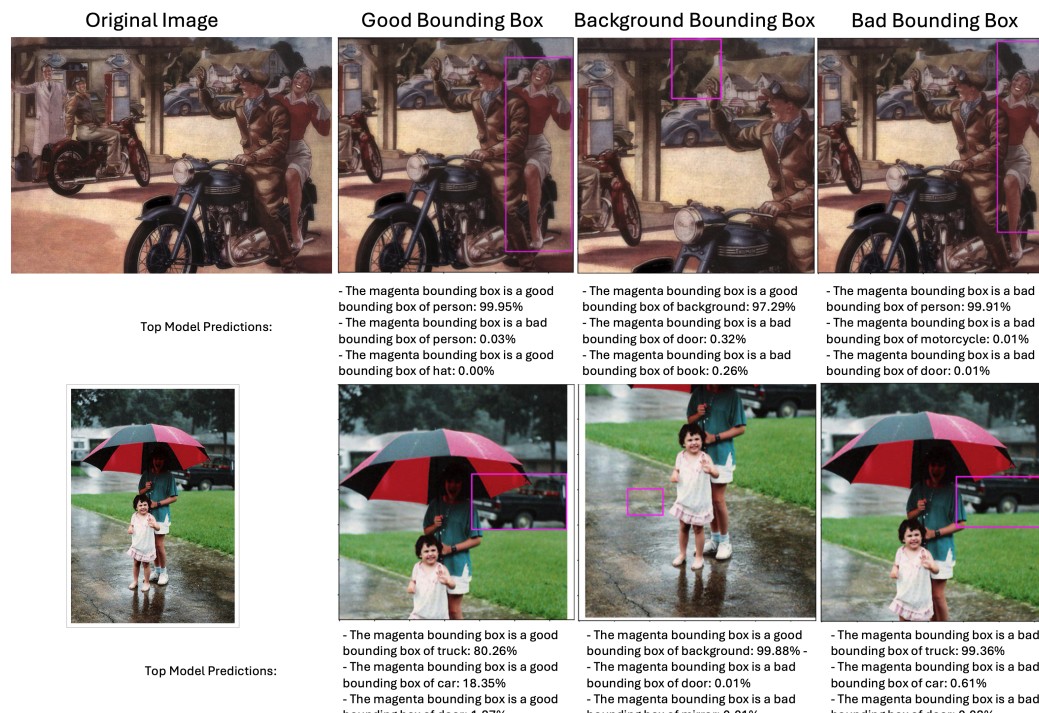

Figure 1: Examples illustrating the three types of generated input images and top model predictions. Each generated image crop contains one magenta bounding box to mark the target object's location.

of bounding box quality, enhancing the overall reliability of pseudo-labels. By screening the quality of pseudo-labels, our approach can reduce accumulated pseudo label errors and contribute to faster convergence and better performance in semi-supervised object detection.

# 3 METHODOLOGY

Our goal is to transform CLIP into a grading model that evaluates both the accuracy of an annotation's object label and the quality of its corresponding bounding box. Below, we detail the task formulation, data preparation, model modification, and training strategy.

## 3.1 TASK FORMULATION

We formalize the grading problem as an image-text matching task using CLIP's contrastive learning framework. However, assessing the quality of bounding box annotations presents several unique challenges that extend beyond typical classification tasks:

**Context Dependency**: The quality of a bounding box is inherently context-dependent. For example, a partially visible object in a bounding box due to occlusion or image framing (e.g. the truck in the $2^{nd}$ example of Figure 1) does not necessarily indicate a poor-quality bounding box. The model must account for these contextual nuances.

**Spatial Reasoning**: The task demands sophisticated spatial reasoning capabilities. The model needs to understand complex spatial relationships, including different viewpoints, occlusions, and potential interactions between objects and/or the background. This requirement significantly surpasses the spatial understanding typically needed for standard image classification tasks.

**Multi-modal Integration**: The task involves integrating both visual and semantic (textual) information. The model must align visual features of the bounding box region with the associated text prompts to determine if the bounding box accurately captures the target object in the text prompt.

To address these challenges, we propose to evaluate each annotation independently within each image, ensuring that the input image to CLIP contains only a single bounding box, represented in magenta to reduce ambiguity, as this color is seldom present in the original images. This configuration allows the model to concentrate on assessing the accuracy and quality of individual bounding box annotations. Furthermore, we utilize an image preprocessor that crops a region surrounding the annotation (see examples in Figure 1), enabling the model to focus specifically on the semantics of the annotation. Text prompts are constructed to describe both the target object and the quality of the bounding box - either "good" or "bad", i.e. for an object detection dataset, we generate corresponding textual prompts for each of the object classes:

• **Good bounding box**: "The magenta bounding box is a good bounding box of [object class]."

• **Bad bounding box**: "The magenta bounding box is a bad bounding box of [object class]."

• **Background box**: "The magenta bounding box is a good bounding box of background."

Each original object class maps to two text prompts, one for high quality bounding box and one for low quality bounding box. This results in a contrastive learning problem with $2n + 1$ prompts for $n$ object classes, where the additional category corresponds to a background class where no target object is in the box.

Let $T = \{t_1, t_2, \ldots, t_{2n+1}\}$ represent the set of text prompts for a dataset. The training goal is to maximize the alignment between the visual encoding $f_v(I)$ of an image crop $I$, and the text encoding $f_t(t_i)$ of all correct text prompts in a batch, vice versa for text embedding. In CLIP's original contrastive learning framework, the ground truth for each batch is an identity matrix, where each image has a one-to-one correspondence with a single text prompt. In contrast, our task allows for multiple correct matches within a batch, similar to (Khosla et al., 2021), as the same (object class, bounding box quality) combination will appear multiple times in a batch. This is formalized as minimizing the two cross-entropy losses $\mathcal{L}_{\text{image}}$ and $\mathcal{L}_{\text{text}}$:

$$\mathcal{L}_{\text{image}} = \frac{1}{|B|} \sum_{I \in B} \left( -\log \frac{\sum_{t^c \in B} \exp(f_v(I) \cdot f_t(t^c))}{\sum_{t \in B} \exp(f_v(I) \cdot f_t(t))} \right) \tag{1}$$

$$\mathcal{L}_{\text{text}} = \frac{1}{|B|} \sum_{t \in B} \left( -\log \frac{\sum_{I^c \in B} \exp(f_v(I^c) \cdot f_t(t))}{\sum_{I \in B} \exp(f_v(I) \cdot f_t(t))} \right) \tag{2}$$

where $t^c$ is the correct textual prompt corresponding to the image, and $I^c$ is one correct image corresponding to the text prompts. For both, there would be multiple matches within a batch, thus the sum. $f_v(.)$ and $f_t(.)$ are image and text embedding respectively, and B are all the images and text prompts in a batch.

To allow for multiple correct matches, we also modify the way the ground truth (GT) matrix is encoded. For each image in a batch, a ground truth vector is created where all correct text prompts in the batch are given a value of 1, and 0 otherwise. Then each GT vector is normalized by its sum to ensure that the total probability mass sums to 1 and then they are stacked together to form the GT matrix of a batch. This modified GT matrix is no longer diagonal as in CLIP, but it is still a symmetric matrix. Using this modified ground truth, we compute the cross-entropy losses defined above for both image-to-text and text-to-image matches with the final loss being the average of these two. The image loss encourages the model to align image embeddings with all correct text embeddings and the text loss encourages the model to align text embeddings with all correct image embeddings, allowing for aligning the model to our grading task.

By framing the problem in this manner, we leverage CLIP's ability to process and align visual and textual information. Simultaneously, we guide the model to learn the nuanced characteristics that distinguish between high-quality and suboptimal bounding boxes across object classes in the dataset.

## 3.2 TRAINING DATA PREPARATION

We use the COCO and LVIS (Large Vocabulary Instance Segmentation) (Gupta et al., 2019) datasets for training and evaluation, both of which provide diverse object categories and bounding box an-

notations. To create a robust dataset for anotation grading, we augment each ground truth bounding box to generate good, bad, and background examples:

• **Good bounding boxes**: The GT annotations provided by the dataset serve as examples of good bounding boxes; however, they are not without imperfections. The quality of a "good" bounding box can vary and may include some level of noise, which deep learning models generally tolerate and overcome. As a side effect, the noise gives the model a sense of intra-class variability which can be beneficial for robust training (Vahdat, 2017).

• **Bad bounding boxes**: These are generated by randomly modifying GT bounding boxes where the position and size deviates from the ground truth with IoU between 0.5 and 0.8, hence the object class label is correct but the bounding box is not snug. We empirically selected this IoU range to ensure that these randomly disturbed bounding boxes are sufficiently challenging for the model to learn.

• **Background bounding boxes**: These are randomly generated bounding boxes that do not have major overlap (i.e. IoU $\leq 20\%$) with any labelled objects in the same image. This is the garbage class - lack of clear target object in the box.

These augmentation steps are crucial for creating a balanced and informative dataset using only standard object detection annotations. It provides the model with a range of examples from accurate bounding boxes to subtly incorrect ones to irrelevant ones. This procedure is used to generate both training and evaluation data using standard COCO and LVIS training and validation splits.

After generating these bounding box variants, a filtering step is applied where we remove objects that have bounding boxes with both height and width smaller than 20 pixels. This selective filtering still retains objects that may be thin or elongated (e.g., a knife or a pole), but removes tiny bounding boxes. This decision is based on several reasons: the difficulty in generating meaningful "bad" examples for extremely small objects, and the often low-quality of image patches for very small objects in COCO and LVIS. This filtering process removes approximately 7% of the COCO dataset, which we deemed an acceptable trade-off between data preservation and training signal quality.

Our image preprocessing pipeline diverges from the original approach used in the CLIP image preprocessor. To encode the spatial relationships of bounding boxes, as illustrated in Figure 1, we draw magenta boxes (3 pixels thick) directly on the image allowing the model to perceive the bounding boxes as integral components of the image. Instead of the center crop used by CLIP, we dynamically crop the image in a square around the bounding box, with the length of the side set to 1.2 to 1.5 times the bounding box size, ensuring enough context is included for spatial reasoning. If the bounding box is near the edge of the image, the crop is adjusted to maximize the inclusion of original image content. Additionally, the center of the crop is randomized to prevent positional biases - the bounding box is not always in the center. Hence this image preprocessing pipeline maintains spatial relationships and context while adapting the bounding-box-added input to CLIP's architecture.

### 3.3 Model Architecture and Training Setup

We use the largest CLIP model (ViT-L/14@336px) provided by OpenAI as our base architecture. In our experiments, we found that model size significantly impacts performance, with the largest CLIP model yielding significantly better results. To enhance generalization, we add dropout layers (dropout rate 0.25) in both the vision and text encoders. Flash Attention 2(Dao, 2024) is used which speeds up the computation and reduces memory bandwidth requirements of attention operations. Other model architecture and parameters remain the same as original CLIP so as to take advantage of the pretrained model. Our primary focus is on developing an effective learning strategy for this new task, as we believe the core CLIP architecture is sufficiently powerful when coupled with the right training strategy.

The dataloader makes a best effort to sample one bad, one background and one good bounding box from each image to ensure class balance. To achieve larger batch sizes, our training pipeline employs mixed precision training, gradient checkpoint and gradient accumulation. The larger batch size helps to provide a diverse set of negative examples in each batch which is important for contrastive learning. The effective batch size is roughly 2500, distributed across 6 Nvidia A100 GPUs.

For optimization, Adam optimizer (Kingma, 2014) and cosine annealing scheduler (Loshchilov & Hutter, 2017) are used. We set the initial learning rate to 5e-5, which we found to provide a good balance between learning speed and stability. The $\beta_1$ and $\beta_2$ parameters of Adam are set to 0.9 and 0.98 respectively, following common practice in CLIP training. Since dropout is added for regularization, no weight decay is used.

## 3.4 LEARNING STRATEGY

To understand the relative importance of the different training techniques in our task, we conducted a series of experiments including prompt engineering, changes in model size, and different fine-tuning strategies i.e. full model fine-tuning versus fine-tuning only the vision encoder or text encoder.

Our results showed that fine-tuning only the vision encoder (Dosovitskiy et al., 2020) yielded performance nearly identical to full model fine-tuning, while fine-tuning only the text encoder led to significantly worse performance. To further investigate the role of the text domain in our model, we conducted additional experiments with prompt engineering. We designed more elaborate prompt mixes that included several detailed descriptions of what constitutes a good or bad bounding box. For example, instead of simply stating "a good bounding box of [object]", we used prompts like "a precise bounding box that tightly encloses the entire [object]" or "an inaccurate bounding box that is too large for [object]". Surprisingly, these more descriptive prompts did not result in faster training or better accuracy compared to our simpler prompt structure. This suggests that the bulk of the learning happens in the visual domain as shown in Section 4.2.

These findings offer valuable insights into the nature of the bounding box quality assessment task. They suggest that future research in this area might benefit from a greater focus on enhancing visual processing capabilities. Despite that, we believe that the combination of visual and textual information in CLIP is also valuable which we discuss further in Section 4.3.

Overall, our modeling approach offers several key innovations and advantages. By modifying the CLIP model and training strategy, we created a model tailored to the nuances of bounding box quality assessment. The result is a flexible and scalable solution that can adapt to the diverse needs of real-world applications, providing a powerful tool to assess and improve the quality of object detection labels.

## 4 EXPERIMENTS AND MAIN RESULTS

**Evaluation Datasets** We conducted a comprehensive set of experiments to evaluate the performance and characteristics of ClipGrader. This section details our experimental setup, metrics, and findings. Our experiments use two widely used object detection datasets: COCO and LVIS. The COCO dataset contains 80 object categories, while LVIS features a much larger set of 1203 categories. For both datasets, we utilized their provided training and validation splits for our experiments. During training, each epoch took approximately 2-3 hours to complete. We ran our experiments for a maximum of 15 epochs, though in many cases, convergence was achieved earlier.

**Evaluation Metrics** While overall classification accuracy serves as a general indicator of model performance, our evaluation also looked at metrics more directly related to our goal of assessing and filtering labels. Specifically, we calculated: (1) Mean recall of good labels: This metric (higher is better) measures the ability to correctly retain high-quality labels and lower the workload of human review. (2) Mean false acceptance (false positives) of bad labels: This metric (lower is better) helps us gauge low-quality label contamination, particularly for cases where the object class is correct but the bounding box is not snug. These metrics align with common types of labeling errors and with our objective of efficiently screening low-quality labels for human review. They are first computed per-class, then the mean across all classes is reported, minimizing the impact of class imbalance.

## 4.1 PERFORMANCE, DATA EFFICIENCY AND GENERALIZATION

In the first set of experiments, we evaluated the performance and data efficiency of our approach on COCO dataset and its ability to generalize to the more complex LVIS dataset. We trained several models on varying amounts of the COCO training set (1%, 5%, 10%, and 100%) and tested on the COCO validation split. A model was also trained and tested on the LVIS dataset (100%), which

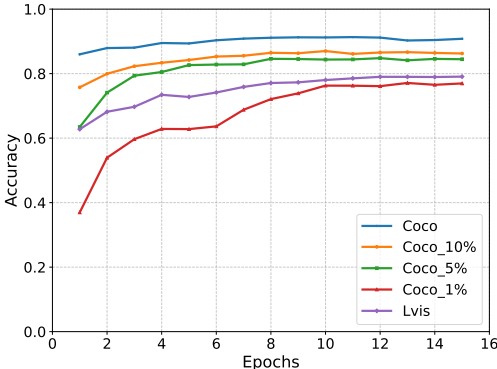

Figure 2: Accuracy vs. Epochs for different training dataset configurations

| Data Config. | Acc. | Mean Recall of Good Labels | Mean False Acceptance of Bad Labels |
|---|---|---|---|
| COCO 100% | 91% | 84.7% | 1.8% |
| COCO 10% | 87% | 73.1% | 2.1% |
| COCO 5% | 84% | 70.5% | 2.5% |
| COCO 1% | 77% | 61.2% | 8.2% |
| LVIS 100% | 79% | 76.2% | 3.9% |

Table 1: Accuracy for different training configurations and example trade-off between mean recall of good labels and false acceptance of bad labels

presents a much more challenging scenario with many more classes and smaller objects. The main results are shown in Figure 2 and Table 1.

These results indicate that ClipGrader can be effectively fine-tuned with a small fraction of the available data, making it particularly suitable for scenarios where labeled data is scarce or expensive to obtain. Figure 2 demonstrates remarkable data efficiency: with just 10% of COCO training data, our model achieves 87% accuracy, compared to 91% when using 100% of the dataset. Table 1 offers more direct insights into our model's performance on the label grading task. With only 10% of the COCO training data, it achieves 73.1% recall of good labels and rejects 97.9% of inaccurate annotations, compared to 84.7% and 98.2% respectively when trained on the full dataset. This suggests that our approach can be effectively bootstrapped with limited seed data and be useful at the early stage of data curation.

As we reduce the amount of training data, we observe a gradual decrease in performance. However, the degradation is not linear. Even with just 5% of the data, the model still maintains 84% accuracy. This robustness to data reduction further underscores the efficiency of our approach. It's worth noting that as the amount of training data decreases, we generally see a worsening in all metrics. This trade-off is most pronounced when using only 1% of the COCO data, where bad label contamination jumps to 8.2%. This suggests that while our model can operate with very limited data, there's a threshold below which performance begins to degrade more rapidly.

The model's generalization capability is evident in its performance on the LVIS dataset. Despite the underlying ML task on LVIS being a 2407-way classification task, compared to COCO's 161-way, the model still achieves approximately 80% accuracy. Its performance on a much larger and more diverse dataset speaks to the generalizability of ClipGrader. These results suggest that ClipGrader is not only data-efficient but also capable of handling complex, multi-class scenarios. In real-world applications, where the number of target classes is likely to be smaller than in LVIS or COCO and object image quality potentially better, we can expect even stronger performance.

Figure 3 illustrates the trade-off between recall of good labels and false acceptance rate of bad labels across different model configurations. The ROC-like curves demonstrate the strong capability of ClipGrader across various datasets and training data quantities. While the raw probabilities output by ClipGrader are not calibrated, the curves indicate that it is relatively straightforward to find an appropriate threshold to balance the recall of good labels against the false acceptance of bad labels. Notably, even with reduced training data (10% or 5% of COCO), the model maintains a reasonable balance between correctly identifying good labels and rejecting bad ones. Moreover, the LVIS curve also exhibits the strong trade-off capability, even though it has a slightly lower area under the curve compared to COCO due to the dataset's increased complexity. This flexibility is particularly desirable, as it allows the model to be tuned to serve different application needs. This adaptability makes ClipGrader a versatile tool for dataset review across various requirements.

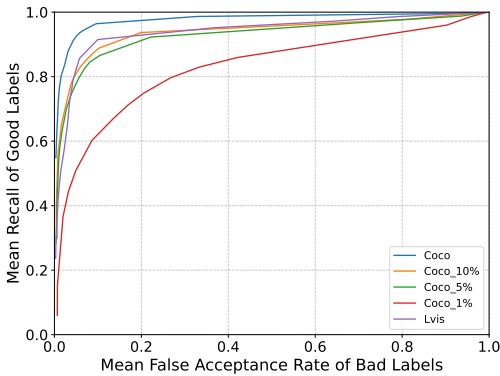 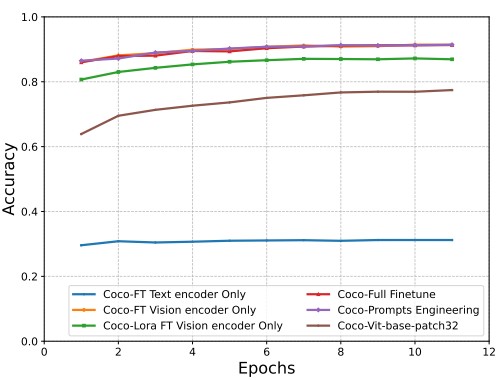

Figure 3: Trade-off between mean recall of good labels and false acceptance of bad labels for different model configurations

Figure 4: Accuracy vs. Epochs for different fine-tuning strategies and prompt engineering

## 4.2 EXPERIMENTAL ZERO-SHOT GRADING

To further explore the generalization capabilities of ClipGrader, we applied the model trained on 100% of COCO directly to grade bounding boxes of unseen classes in the LVIS dataset. This challenging task required the model to assess bounding boxes for over 1100 unseen classes during training (we removed the common classes between COCO and LVIS), relying solely on the pretraining knowledge of CLIP and the concept of bounding boxes learned during our fine-tuning process.

In this evaluation, the ClipGrader model, trained on the COCO dataset, demonstrates a constrained yet significant ability to assess bounding box quality across over 1100 entirely unseen object classes - it achieved an accuracy of approximately 11%. As can be expected, many unseen class samples are classified as background which makes sense since they are indeed "background" during its training on COCO. While this accuracy is considerably lower than the model's performance on seen classes, it should be noted that it significantly outperforms random guessing in over 1100 categories. Moreover, ClipGrader's zero shot capability is weaker than CLIP's capability for classification, most likely due to the limitation of fine tuning dataset and the newly added 'background' category.

The relatively low accuracy suggests that this zero-shot capability is not yet sufficient for practical applications. Some amount of labeled bounding box data for new classes of interest is still necessary for the model to achieve high-quality assessment and seperate them from the 'background' class. However, this result provides evidence that our fine-tuning process teaches the model to learn a generalizable concept of bounding box quality, which lays the foundation for data efficient fine tuning or few-shot learning for new classes.

## 4.3 FINE-TUNING STRATEGIES AND MODEL COMPONENT ANALYSIS

To understand the relative importance of different components in the CLIP model and the effectiveness of various fine-tuning strategies, we conducted a series of experiments. Figure 4 provides the results of this comprehensive study.

We first investigated the impact of model size by comparing our default model (clip-vit-large-patch14-336) with a smaller variant (clip-vit-base-patch32). The smaller model, approximately one-third the size of our default, showed a significant performance drop when both models were trained on the full COCO dataset. Accuracy decreased by about 12%, with the smaller model's overall performance comparable to the larger model trained on only 1% of the COCO dataset. This experiment underscores the importance of model capacity for the bounding box assessment task.

To disentangle the contributions of the vision and language components, we compared several targeted fine-tuning strategies: fine-tuning only the Vision encoder, LORA (rank 32) fine-tuning (Hu et al., 2021) of only the Vision encoder, fine-tuning only the Text encoder, full model fine-tuning, and prompt engineering during full model fine-tuning. Our results reveal the crucial role of the Vision encoder in this task. Fine-tuning only the Vision encoder yielded performance nearly identical

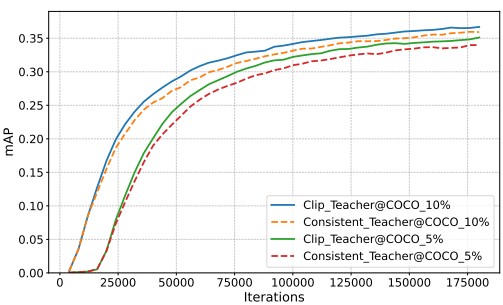

| Dataset | Method | $AP_{50}$ | $AP_{75}$ | $AP_{50:95}$ |
|---------|--------|-----------|-----------|--------------|
| 5% | Consistent | 49.30 | 36.14 | 33.98 |
|    | **Clip** | **50.76** | **37.52** | **35.11** |
| 10% | Consistent | 51.37 | 38.41 | 35.92 |
|     | **Clip** | **52.53** | **39.22** | **36.69** |

Table 2: Performance comparison of Clip-Teacher and Consistent-Teacher. when using 5% or 10% labeled COCO dataset.

Figure 5: Clip-Teacher vs. Consistent-Teacher

to full model fine-tuning, both achieving above 91% accuracy. LORA fine-tuning the Vision encoder only trains less than 1.5% of the model parameters, and it achieves 87% accuracy. This suggests that bounding box quality assessment relies almost exclusively on visual cues processed by the Vision encoder.

In contrast, fine-tuning only the Text encoder resulted in accuracy stagnating around 30%. This further confirms that bounding box quality assessment is predominantly a visual task, with limited reliance on textual information. Our experiments with more elaborate prompts (labeled as "COCO-Prompts Engineering" in Figure 4) demonstrated that additional prompt engineering does not improve performance compared to simple prompts. This finding aligns with our observation of the task's visual nature and the limited role of textual inputs.

These results have important implications for the design and training of bounding box quality assessment models. Computational resources and efforts to improve performance should give higher priority to the model's visual processing capabilities rather than refining textual inputs or text processing components. As noted earlier, however, we believe that while the task is primarily visual, the interplay between visual and textual information in the CLIP architecture still contributes to its overall effectiveness in ways that are not immediately apparent. This can also be seen from the prior results where the fine tuned model still exhibits limited but non-trivial zero-shot grading capability for unseen classes.

## 4.4 CASE STUDY IN SEMI-SUPERVISED OBJECT DETECTION: CLIP-TEACHER

To showcase a simple practical utility of ClipGrader, we integrate it into a semi-supervised object detection (SSOD) framework, referred to as "CLIP-Teacher". Pseudo-labeling is essential in SSOD, where bounding boxes generated by a teacher model serve as pseudo-ground truth (pseudo-GT) for training the student model. However, without effective quality control, poor pseudo-labels can degrade overall model performance. By incorporating ClipGrader to screen and retain only high-quality pseudo-labels, CLIP-Teacher improves the effectiveness of the SSOD process.

We implemented CLIP-Teacher by integrating ClipGrader into the Consistent-Teacher SSOD framework. We compare our method against the original strategy used in Consistent-Teacher on partially labeled CoCo datasets. Two setups are tested (5% and 10% percent of the COCO training dataset) in our comparison. The only difference between CLIP-Teacher and Consistent-Teacher is the use of ClipGrader to filter the pseudo-labels generated by the teacher model. Both models were trained on the same partially labeled dataset (5% or 10% of COCO dataset) and use the same warm up check-point(12k iterations) to ensure a fair comparison. Importantly, ClipGrader was also trained using the same 5% or 10% of COCO labeled data to avoid data leakage. While ClipGrader is not trained with any image data augmentation, examples in Fig. 6 (Appendix A) shows it is quite robust to the strong data augmentation employed by Consistent Teacher. Please refer to Appendix A for more implementation details.

Figure 5 shows the comparison of mAP (mean Average Precision) performance between the two approaches across training iterations. The results demonstrate that CLIP-Teacher consistently outperforms Consistent-Teacher after warm up, particularly by filtering out low-quality pseudo-labels that the baseline model does not eliminate. This results in faster convergence and improved model

accuracy throughout the training process. The performance metrics in Table 2 further confirm the advantage of CLIP-Teacher, with an improvement in AP50, AP75, and AP50:95 across the board. These results highlight ClipGrader's capability to serve as a robust quality control mechanism, ensuring that only higher quality pseudo-labels are used in training, ultimately leading to more accurate object detection models in semi-supervised settings.

## 5 DISCUSSION AND FUTURE WORK

An important direction for future work is the application of ClipGrader to improve existing datasets through error analysis and correction. Our examination of mispredictions on the COCO test set reveals that in many cases where ClipGrader disagrees with ground truth labels, the discrepancy is due to label ambiguity or errors in the original annotations. Figures 7 and 8 (in Appendix B) illustrate several such examples where ClipGrader's judgments appear more accurate or nuanced than the original COCO annotations. This finding suggests that ClipGrader has learned a robust concept of bounding box quality that can generalize beyond the errors present in COCO. By flagging ambiguous or potentially incorrect annotations for human review, we could create a feedback loop to continually improve dataset quality. This process could be particularly valuable for large-scale datasets like COCO, where comprehensive manual review is impractical.

Another future work could focus on developing efficient workflows that integrate ClipGrader into the dataset curation process, allowing for targeted human intervention to resolve ambiguities and correct errors identified by the model. This could involve creating interfaces that present flagged annotations to human annotators along with ClipGrader's predictions, facilitating quick decision-making and corrections. In the early stages of data collection and annotation, ClipGrader can be bootstrapped with a small amount of seed data to guide the rest of the curation process. Additionally, studying the types of errors and ambiguities frequently identified by ClipGrader could provide insights into common pitfalls in annotation processes, informing the development of improved annotation guidelines and quality control measures for future data collection efforts.

Several avenues for future research could further enhance our approach. Extending the method to grade multiple bounding boxes in a single image simultaneously would bring higher efficiency. It's worth noting that while ClipGrader can assess existing annotations, it is not trained to find missing annotations in an image, presenting an area for potential future work. While we focused on bounding box quality for object detection, similar approaches could be developed for other computer vision tasks, such as segmentation. Furthermore, our methodology could be applied to larger, more advanced vision-language models, e.g. Florence-2(Xiao et al., 2024), which could lead to even stronger results given its increased capacity and better understanding of visual/textual relationships.

## 6 CONCLUSION

In this paper, we introduced a novel CLIP-based approach for assessing bounding box quality in object detection datasets. Our method demonstrates high effectiveness and efficiency across diverse object classes and dataset complexities. It maintains strong performance even with limited data, achieving 87% accuracy using only 10% of COCO. This data efficiency promises to automate and improve annotation quality control from initial collection to ongoing dataset refinement.

ClipGrader's generalization capabilities also extend to complex datasets like LVIS, highlighting its adaptability. Our experiments reveal the primacy of visual processing for this task, informing future model designs. This work represents an advancement in AI assisted data curation and verification for object detection, contributing to enhanced reliability and performance of object detection systems across various requirements.

## 7 REPRODUCIBILITY STATEMENT

To make the experiments and models reproducible, the train/testing data and the benchmark details are provided in Section 3 and Section 4. Example inference code and model checkpoint will be released on the ClipGrader project GitHub page.

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

## A    MORE DETAILS ON CLIP-TEACHER

CLIP-Teacher is a simple extension of Consistent-Teacher, with an additional pseudo-label screening step. After the teacher model generates pseudo-labels (object class and bounding box) for the unlabeled images in a batch, CLIP-Teacher uses ClipGrader to filter out low-quality pseudo-labels.

In CLIP-Teacher, each pseudo-label, along with the corresponding image, is first passed to the image preprocessor described in Sec. 3.1. The image crop, containing the bounding box and surrounding region, is then evaluated by ClipGrader for quality. A bounding box is considered high-quality if the predicted class label matches the pseudo-label and the probability score for the corresponding text prompt of "good" bounding box exceeds a predefined threshold.

In this pipeline, ClipGrader acts as an additional quality control mechanism, ensuring that only high-quality pseudo-labels are selected for student model training, while low-quality ones are discarded. Examples of ClipGrader's assessments are shown in Figure 6. Notably, even though ClipGrader is not trained with image augmentations, it remains robust to the strong augmentations, e.g. Random-Resize, RandErase, (Xu et al., 2021) used in Consistent-Teacher pipeline.

To ensure consistency with prior art, we used the default settings of Consistent-Teacher (Wang et al., 2023), using RetinaNet (Lin et al., 2020) as the object detector based on MMdetection framework (Chen et al., 2019). For this experiment, we tried two setups, using 5% or 10% CoCo training data respectively. Both Consistent-Teacher and Clip-Teacher models were trained on the same 5% or 10% of COCO dataset and use the same warm up checkpoint (12k iterations) to ensure a fair comparison. Importantly, ClipGrader used by CLIP-Teacher is also trained using the corresponding 5% or 10% of COCO data to avoid data leakage. Both CLIP-Teacher and Consistent-Teacher are optimized using SGD, utilizing a constant learning rate of 0.01, a momentum of 0.9, and a weight decay of 0.0001. Both CLIP-Teacher and Consistent-Teacher are trained for 52,000 iterations across 4 Nvidia A100 GPUs. During training, Consistent-Teacher generates an average of approximately 4

pseudo-labels per image, while CLIP-Teacher generates an average of 3.25 pseudo-labels per image, filtering out approximately 19% of the pseudo-labels in our current implementation.

| Incorrect Label | Accurate Label
Low Quality BBox | Accurate Label
High Quality BBox |
|---|---|---|

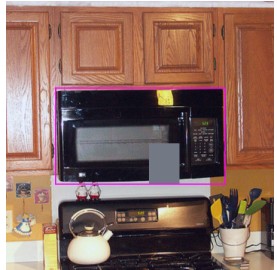 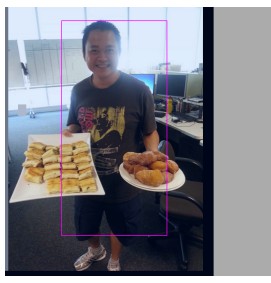 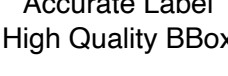 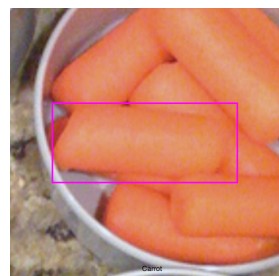

Pseudo Label: tv
ClipGrader: microwave

Pseudo Label: person
ClipGrader: person

Pseudo Label: carrot
ClipGrader: carrot

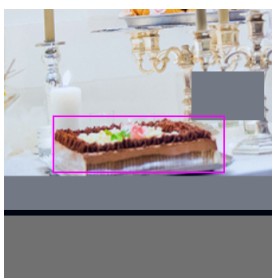 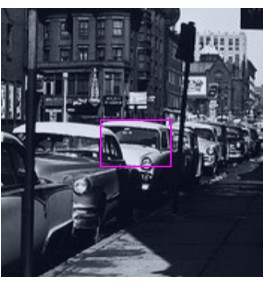 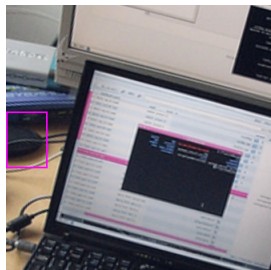

Pseudo Label: pizza
ClipGrader: cake

Pseudo Label: car
ClipGrader: car

Pseudo Label: mouse
ClipGrader: mouse

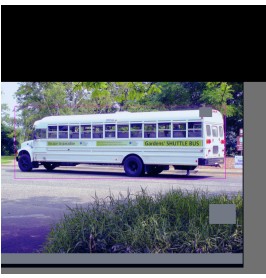 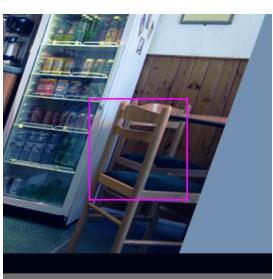 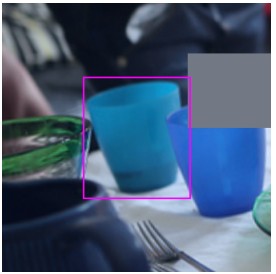

Pseudo Label: truck
ClipGrader: bus

Pseudo Label: chair
ClipGrader: chair

Pseudo Label: cup
ClipGrader: cup

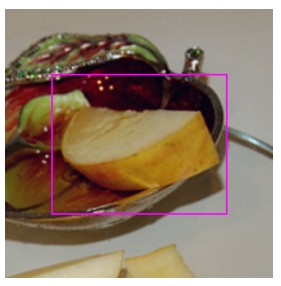 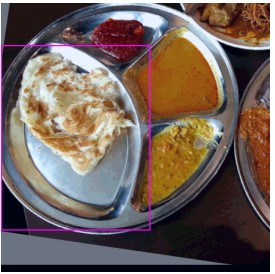 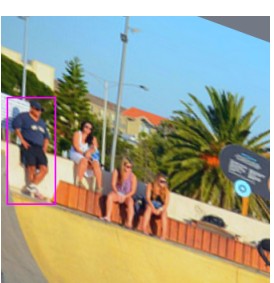

Pseudo Label: donut
ClipGrader: apple

Pseudo Label: bowl
ClipGrader: bowl

Pseudo Label: person
ClipGrader: person

Figure 6: Example of ClipGrader's assessment on pseudo labels proposed by Consistent-Teacher

# B    EXAMPLES FROM ERROR ANALYSIS

GT: motorcycle
Top predictions:
The magenta bounding box is a bad bounding box of motorcycle: 97.31%
The magenta bounding box is a good bounding box of motorcycle: 2.67%
The magenta bounding box is a bad bounding box of bicycle: 0.01%

GT: cake
Top predictions:
The magenta bounding box is a good bounding box of sandwich: 99.46%
The magenta bounding box is a good bounding box of cake: 0.30%
The magenta bounding box is a bad bounding box of sandwich: 0.11%

GT: chair
Top predictions:
The magenta bounding box is a bad bounding box of bench: 94.14%
The magenta bounding box is a bad bounding box of desk: 2.95%
The magenta bounding box is a bad bounding box of chair: 2.84%

GT: banana
Top predictions:
The magenta bounding box is a bad bounding box of banana: 99.98%
The magenta bounding box is a good bounding box of banana: 0.01%
The magenta bounding box is a bad bounding box of apple: 0.00%

GT: cow
Top predictions:
The magenta bounding box is a good bounding box of sheep: 84.96%
The magenta bounding box is a bad bounding box of sheep: 14.98%
The magenta bounding box is a good bounding box of cow: 0.03%

GT: clock
Top predictions:
The magenta bounding box is a good bounding box of tv: 99.43%
The magenta bounding box is a good bounding box of clock: 0.25%
The magenta bounding box is a bad bounding box of tv: 0.17%

Figure 7: Error analysis examples demonstrating ClipGrader's ability to identify potential labeling errors and ambiguities in the COCO dataset. The images show examples from the COCO test set with ground truth labels and bounding boxes. ClipGrader's top 3 predictions are shown for each example, often revealing more accurate judgments than the ground truth COCO annotations.

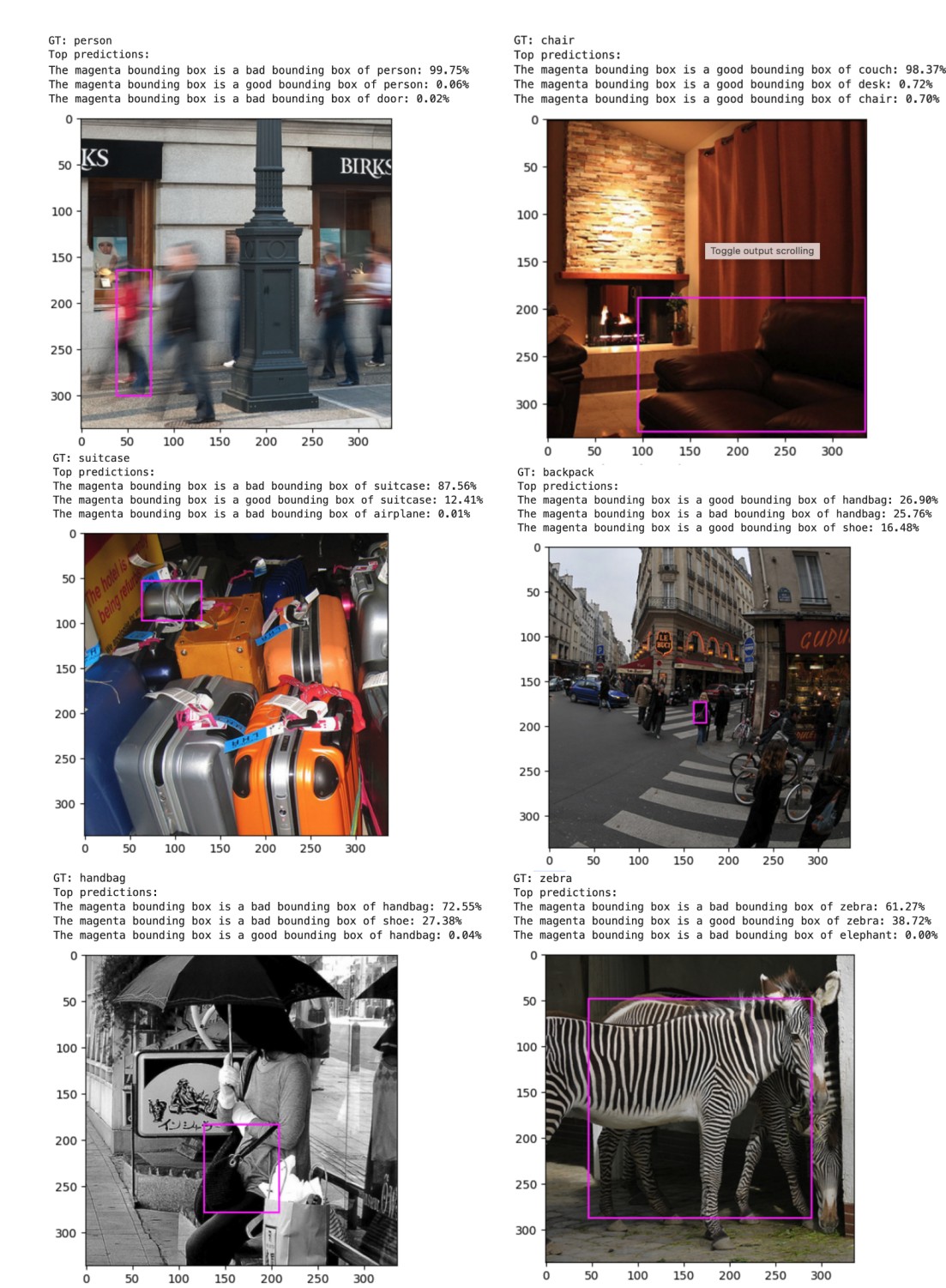

Figure 8: More error analysis examples demonstrating ClipGrader's ability to identify potential labeling errors and ambiguities in the COCO dataset. The images show additional examples from the COCO test set, further illustrating cases where ClipGrader's predictions provide insights into label quality and potential improvements for COCO dataset.

