# OpenReview forum: "ClipGrader: Leveraging Vision-Language Models for Robust Label Quality Assessment in Object Detection"
_ICLR.cc/2025/Conference — Submitted to ICLR 2025_

### Official Review · Reviewer_UE2r · 2024-10-31

**Soundness:** 3
**Presentation:** 2
**Contribution:** 2
**Rating:** 3
**Confidence:** 5

**Summary:**

High-quality annotations are essential for object detection tasks. This paper propose to leverage vision-language models (e.g. CLIP) to automatically assess the accuracy of bounding box annotations. The author tried a lot of ways, including prompt engineering, changes in model size, and different model fine-tuning strategies. The final results demonstrate that the proposed approach can identify errors in existing COCO annotations, highlighting its potential for dataset refinement.

**Strengths:**

1. This paper leverages vision-language models for label quality assessment is valuable.
2. The experimental results show the potential ability for dataset refinement.

**Weaknesses:**

1. Employing pre-trained models to relabel or denoise dataset is not novel. A large amount of literature has demonstrated the ability of multimodal models.
2. The contribution of this paper is limited.
3. The experimental results are not novel. As the CLIP model has been trained on a large number of text-image datasets, including the COCO dataset used in this experiment. I think this paper is more of a good attempt in engineering, leaning towards a technical report.

**Questions:**

See the Weaknesses.

#

---

### Official Review · Reviewer_BMBP · 2024-10-31

**Soundness:** 2
**Presentation:** 3
**Contribution:** 1
**Rating:** 3
**Confidence:** 5

**Summary:**

Proposes CLIPGrader, an approach to fine-tune CLIP to judge the quality (correctness of box position and label) of detection bounding boxes.The approach is shown to achieve high accuracy at label assessment on COCO and LVIS and shown to improve the performance of semi-supervised object detection methods.

**Strengths:**

– The paper is well-written and easy to follow

– The proposed method seems to be quite effective at assessing label quality

**Weaknesses:**

– The paper makes limited technical contributions. It’s main contribution – empirically showing that CLIP can be fine-tuned to assess label quality – is interesting but in my opinion not substantial. The proposed finetuning strategy is a straightforward generalization of the original CLIP objective (and bears similarity to the supervised contrastive loss [A], which the paper should cite).

– The motivation of the paper is somewhat weak. Why is CLIP well-suited to label assessment, beyond being a popular multimodal model? Why not use a specialized approach like RegionCLIP [B], which has been designed with detection in mind?

– The problem formulation is also somewhat contrived. Why treat label quality assessment as a classification problem rather than regressing to the correct coordinates (or a delta therefrom)? Regression would alleviate the need to arbitrarily define “good” and “bad” bounding boxes, and allow for more fine-grained metrics (like AP).

– The paper primarily measures accuracy based on test-set label assessment accuracy. A far more helpful measure would be performance of detection methods that factor in the predicted label quality (maybe by loss weighting, or pseudolabel filtering). While the paper does include a single experiment  in Sec 4.4 on semi-supervised object detection, I think a comprehensive set of additional experiments is required to verify that the proposed task and model is actually useful for a real-world task.

– The paper studies a synthetic label assessment setup, as the “bad” bounding boxes are generated by randomly perturbing bounding boxes. While this is reasonable as a starting point, the paper would be strengthened with experiments on “in the wild” datasets (eg. by having humans annotate ground truth boxes in an existing evaluation set as “good” and “bad”). This is particularly important since prior work has shown that labeling errors in detection are not always random and in-fact can be class and annotation protocol dependent [C].

– The dataset contains examples of good and bad bounding boxes for each class as well as background boxes, but does not include examples of good bounding boxes but for the wrong class? How does the translate to the model’s performance?

[A] Khosla, Prannay, et al. "Supervised contrastive learning." NeurIPS 2020

[B] Zhong, Yiwu, et al. "Regionclip: Region-based language-image pretraining." CVPR 2022

[C] Liao, Yuan-Hong, et al. "Transferring Labels to Solve Annotation Mismatches Across Object Detection Datasets." ICLR 2024

**Questions:**

See weaknesses above for a detailed list of questions.

---

### Official Review · Reviewer_u55H · 2024-11-02

**Soundness:** 3
**Presentation:** 3
**Contribution:** 3
**Rating:** 6
**Confidence:** 4

**Summary:**

The paper introduces ClipGrader, a framework for automatically assessing bounding box quality in object detection datasets by adapting CLIP, a vision-language model, to evaluate label accuracy and spatial alignment of bounding boxes. Evaluation on COCO and LVIS datasets demonstrates the effectiveness of ClipGrader.

**Strengths:**

1. The paper is well-organized and easy to follow, different components are illustrated in detail, making the framework comprehensible and logically structured.
2. Adapting CLIP to assess bounding box quality is innovative and addresses a real challenge in maintaining large-scale object detection datasets. This repurposing of CLIP as a “grader” rather than a classifier or detector is novel and promising.

**Weaknesses:**

1. While the ablation studies (Section 4.3) are detailed, adding a quantitative comparison with simpler baseline methods, such as non-CLIP-based label grading techniques or direct confidence-based bounding box assessments, would strengthen the claim of ClipGrader’s superiority.
2. As mentioned in the introduction (Section 1), widely used datasets such as COCO are subject to label errors and ClipGrader can be used to assess object detection annotations, it would be better to use ClipGrader to filter incorrect annotations in COCO training set and show some performance gains on the test set to validate the usefulness of the proposed method.
3. It’s mentioned in Section 3.3 that “we found that model size significantly impacts performance, with the largest CLIP model yielding significantly better results”, it would be better to have some quantitative comparison between different sizes of CLIP models.
4. The majority of evaluations are conducted on datasets with well-defined and distinct classes (COCO, LVIS). Testing ClipGrader on more complex datasets, such as OpenImages, where bounding box quality varies more, would be better to show its generalizability.

**Questions:**

The idea to utilize CLIP to improve some downstream tasks like object detection is interesting, and this paper is overall good. It would be better to add some baseline results for more comprehensive comparison and validate the proposed method on more complex datasets.

---

### Official Review · Reviewer_9kcp · 2024-11-03

**Soundness:** 2
**Presentation:** 2
**Contribution:** 2
**Rating:** 3
**Confidence:** 3

**Summary:**

This paper introduces ClipGrader, a novel framework that leverages vision-language models to automatically assess the accuracy of bounding box annotations in object detection datasets. It employs CLIP (Contrastive Language-Image Pre-training) to evaluate both class label correctness and spatial precision of bounding boxes, offering a scalable solution for enhancing annotation quality control. ClipGrader demonstrates high accuracy and robustness, achieving 91% accuracy on COCO with a 1.8% false positive rate, and effectively maintaining performance even when trained on a subset of the data. ClipGrader's can help downstream object detection tasks.

**Strengths:**

In general, the proposed method is somehow simple, also the author claimed that the proposed method can be helpful in downstream tasks, which can benefit the process of downstream object detection learning.

**Weaknesses:**

However, I think the proposed method has severe drawbacks as follows:

1. The proposed method is simple and lack novelty. The author only applies a simple classification strategy and a simple contrastive learning pipeline. It is just a simple application of a pre-trained CLIP model. Neither the author proposes a new paradigm (contrastive learning with multiple positive pairs is a common practice in previous papers, especially object detection paper), nor the author has proposed new arch/algorithm to train a grader.

2. For object detection datasets, the author only shows performances on COCO and LVIS datasets. COCO and LVIS share the same data sources. Then the performance, even the few-shot performance doesn't convince me here. The author could show results on downstream object detection benchmarks rather than the COCO source, for example, OpenImages and some small datasets to verify the effectiveness of the proposed method. i.e., on VOC dataset and some auto-matic driving datasets like video surveillance datasets.

3. For the downstream SSOD teacher, though the CLIPGrader is also trained on 10% of the COCO data, CLIP is trained on multiple data sources, which can not prevent the leakage of the data. The author could find better ways to verify the effectiveness of CLIPGrader, i.e., find the noisy annotations in COCO (with ratio and visualization).

**Questions:**

Based on the weaknesses, I have the following questions?

Could the proposed CLIPGrader achieves performance gain on other detection datasets rather than COCO/LVIS?

Will the performance gain mostly from CLIP, rather than the proposed strategy? If so, applying a modern light-weight VLM will be better?

---

### Official Review · Reviewer_2ZSA · 2024-11-03

**Soundness:** 3
**Presentation:** 4
**Contribution:** 3
**Rating:** 6
**Confidence:** 5

**Summary:**

The paper proposes to re-purpose CLIP to evaluate object detection label qualities. CLIP is firstly introduced to align image-level semantics and image captions. On the contrary, this papers leverage visual prompt to promote awareness in certain image regions. The experimental results show that the CLIP-grader achieve non-trivial performances even with 1% COCO data.

**Strengths:**

- The idea to use visual prompts in CLIP to evaluate detection labels is novel
- The performances with large enough training data sounds strong (low false positive rates)
- When using CLIP-grader to improve pseudo-labels, the performances improvements persists. It is non-trivial to translate the performances improvements from labels to model performances in a data-centric manner.

**Weaknesses:**

- Lack of baselines of CLIP-grader when evaluated recall and false positive rates (Table 1)
- Lack of deeper analysis of the tail classes and small bounding boxes, which are considered much more important in object detection.
- Limited zero-shot performances in Sec. 4.2

**Questions:**

- The process of synthesizing unrealistic bounding boxes needs more clarification. What defines a realistic distribution of incorrect boxes? Even preliminary insights would be valuable.
- While CLIPGraders shows promise in evaluating pseudo-labels, can it also improve noisy human annotations? A small-scale study exploring this application would strengthen the paper.

---

### Meta-Review · Area_Chair_fGtJ · 2024-12-24

**Metareview:**

The paper presents ClipGrader, a method leveraging vision-language models, specifically CLIP, to assess the quality of labels in object detection datasets. It evaluates the correctness of class labels and the spatial precision of bounding boxes, demonstrating its potential to refine datasets like COCO and LVIS. Additionally, ClipGrader's integration into semi-supervised object detection pipelines suggests practical benefits in improving pseudo-label quality.

Reviewers acknowledged the idea of adapting CLIP for label quality assessment as interesting and noted its scalability and practical implications for dataset refinement. However, concerns were raised about the lack of technical novelty, as the approach mainly repurposes CLIP without substantial innovation. The experiments were limited to datasets that share overlapping sources, restricting the method's generalizability. Reviewers also highlighted the absence of comparisons with alternative baseline methods and real-world "in-the-wild" error scenarios, questioning the robustness and broader applicability of the method. The authors did not submit a rebuttal during the response phase.

Considering these factors, the AC recommends rejection. While the method demonstrates promise in leveraging pre-trained models for dataset refinement, the lack of response and unaddressed concerns about technical contributions and experimental validation limit its impact and readiness for acceptance.

**Additional Comments On Reviewer Discussion:**

Please refer to the meta review

---

### Decision · Program_Chairs · 2025-01-22

Reject